# Critical Sampling for Data-Driven Modeling of Unknown Dynamical Systems

**Ce Zhang[1]**    **Siqi Wu[2]**    **Zhihai He[3]**    **Zhao Joy Sun[4]**

[1]Carnegie Mellon University    [2]University of Missouri
[3]Southern University of Science and Technology    [4]Hampton University
cezhang@cs.cmu.edu, siqiwu@mail.missouri.edu
hezh@sustech.edu.cn, zhao.sun@hamptonu.edu

**Abstract:** When a robot is exploring an unknown dynamical system, we often face the following important question: what is the minimum number of samples needed for effective learning of its governing laws and accurate prediction of its future evolution behavior, and how to select these critical samples? In this work, we propose to explore this problem based on a design approach. Starting from a small initial set of samples, we adaptively discover critical samples to achieve increasingly accurate learning of the system evolution. We establish a multi-step reciprocal prediction network where forward and backward evolution networks are designed to learn the temporal evolution behavior in the forward and backward time directions, respectively. Very interestingly, we find that the desired network modeling error is highly correlated with the multi-step reciprocal prediction error, which can be directly computed from the current system state. This allows us to perform a dynamic selection of critical samples from regions with high network modeling errors for dynamical systems. Our extensive experimental results demonstrate that our proposed method is able to dramatically reduce the number of samples needed for effective learning and accurate prediction of evolution behaviors of unknown dynamical systems by up to hundreds of times.

**Keywords:** Critical Sampling, Lifelong Learning, Dynamical Systems

## 1   Introduction

When a robot is exploring an unknown dynamical system, such as aerodynamic, climate, or fluid dynamic systems, it needs to collect sensor data samples to learn, model, and predict the behavior of the dynamic system. The behaviors of dynamical systems in the physical world are governed by their underlying physical laws [1, 2]. In many areas of science and engineering, ordinary differential equations (ODEs) and partial differential equations (PDEs) play important roles in describing and modeling these physical laws [3, 4, 5, 6, 7, 8]. Recently, learning-based methods for complex and dynamic system modeling have become an important area of research in machine learning [4, 9, 10]. There are two major approaches that have been explored. The first approach typically tries to identify all the potential terms in the unknown governing equations from a priori dictionary, which includes all possible terms that may appear in the equations [3, 11, 12, 4, 5, 13, 14, 15, 16]. The second approach for data-driven learning of unknown dynamical systems is to approximate the evolution operator of the underlying equations, instead of identifying the terms in the equations [8, 17, 18, 19].

Many existing data-driven approaches for learning the evolution operator typically assume the availability of sufficient data, and often require a large set of measurement samples to train the neural network, especially for high-dimensional systems. For example, to effectively learn a neural network model for the 2D Damped Pendulum ODE system, existing methods typically need more than 10,000 samples to achieve sufficient accuracy [8, 17]. This number increases dramatically with the dimensions of the system. For example, for the 3D Lorenz system, the number of needed samples

7th Conference on Robot Learning (CoRL 2023), Atlanta, USA.

used in the literature is often increased to one million. We recognize that, in practical dynamical systems, such as ocean, cardiovascular and climate systems, it is very costly to collect observation samples. This leads to a new and important research question: *what is the minimum number of samples needed for robust learning of the governing laws of an unknown system and accurate prediction of its future evolution behavior?*

In this work, we propose a critical sampling scheme for accurately learning the evolution behaviors of unknown dynamical systems. We start with a small set of initial samples, then iteratively discover and collect critical samples to obtain more accurate network modeling of the system. During critical sampling, the basic rule is to select the samples from regions with high network modeling errors so that these selected critical samples can maximally reduce the overall modeling error. However, the major challenge here is that we do not know network modeling error. To address this challenge, we establish a multi-step reciprocal prediction framework where a forward evolution network and a backward evolution network are designed to learn and predict the temporal evolution behavior in the forward and backward time directions, respectively. Our hypothesis is that, if the forward and backward prediction models are both accurate, starting from an original state $A$, if we perform the forward prediction for $K$ times and then perform the backward prediction for another $K$ times, the final prediction result $\bar{A}$ should match the original state $A$. The error between $\bar{A}$ and $A$ is referred to as the *multi-step reciprocal prediction error*.

Very interestingly, we find that the network modeling error is correlated with the multi-step reciprocal prediction error. Note that multi-step reciprocal prediction error can be directly computed from the current system state, without the need to know the ground-truth system state. This allows us to perform a dynamic selection of critical samples from regions with high network modeling errors and develop an adaptive learning method for dynamical systems. Our extensive experimental results demonstrate that our proposed method is able to dramatically reduce the number of samples needed for effective learning and accurate prediction of evolution behaviors of unknown dynamical systems.

## 2 Method

### 2.1 Problem Formulation

In this work, we focus on learning the evolution operator $\mathbf{\Phi}_\Delta : \mathbb{R}^n \to \mathbb{R}^n$ for autonomous dynamical systems, which maps the system state from time $t$ to its next state at time $t + \Delta$: $\mathbf{u}(t + \Delta) = \mathbf{\Phi}_\Delta(\mathbf{u}(t))$. It should be noted that, for autonomous systems, this evolution operator $\mathbf{\Phi}_\Delta$ remains invariant over time. It only depends on the time difference $\Delta$. For an autonomous system, its evolution operator completely characterizes the system evolution behavior [8, 17, 20].

Our goal is to develop a deep neural network method to accurately learn the evolution operator and robustly predict the long-term evolution of the system using a minimum number of selected critical samples. Specifically, to learn the system evolution over time, the measurement samples for training the evolution network are collected in the form of pairs. Each pair represents two solution states along the evolution trajectory at time instances $t$ and $t + \Delta$. For simplicity, we assume that the start time is $t = 0$. Using a high-accuracy system solver, we generate $J$ system state vectors $\{\mathbf{u}^j(0)\}_{j=1}^J$ at time 0 and $\{\mathbf{u}^j(\Delta)\}_{j=1}^J$ at time $\Delta$ in the computational domain $D$. Thus, the training samples are

$$\mathcal{S}_F = \{[\mathbf{u}^j(0) \to \mathbf{u}^j(\Delta)] : \mathbf{u}^j(0), \mathbf{u}^j(\Delta) \in \mathbb{R}^n, 1 \le j \le J\}. \tag{1}$$

It is used to train the forward evolution network $\mathcal{F}_\theta$ which approximates the forward evolution operator $\mathbf{\Phi}_\Delta$.

### 2.2 Multi-Step Reciprocal Prediction Error and Critical Sampling

In this section, we show that there is a strong correlation between the multi-step reciprocal prediction error and the network modeling error of the temporal evolution network $\mathcal{F}_\theta^m$.

**(1) Multi-step reciprocal prediction.** In our multi-step reciprocal prediction scheme, we have a forward temporal evolution network $\mathcal{F}_\theta^m$ and a backward evolution network $\mathcal{G}_\vartheta^m$, which model

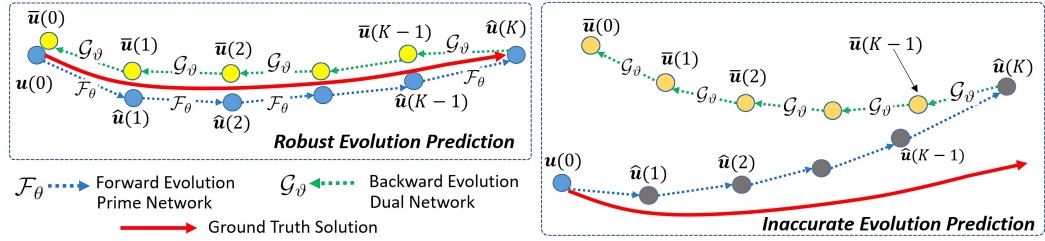

Figure 1: Illustration of the proposed idea of multi-step reciprocal prediction error.

the system evolution behaviors in the forward and backward time directions. The forward and backward evolution networks, $\mathcal{F}_\theta$ and $\mathcal{G}_\vartheta$, allow us to iteratively predict the system's evolution in both forward and backward directions. As illustrated in Figure 1, if the forward and backward evolution networks $\mathcal{F}_\theta^m$ and $\mathcal{G}_\vartheta^m$ are both well-trained, accurately approximating the forward and backward evolution operators, for an arbitrarily given system state $\mathbf{u}(0)$, the one-step reciprocal prediction error $\mathbb{E}[\mathbf{u}(0)] = \|\mathbf{u}(0) - \bar{\mathbf{u}}(0)\| = \|\mathbf{u}(0) - \mathcal{G}_\vartheta^m[\mathcal{F}_\theta^m[\mathbf{u}(0)]]\|$ should approach zero. Now, we extend this one-step reciprocal prediction to $K$ steps. As illustrated in Figure 1, starting from the initial condition $\mathbf{u}(0)$, we repeatedly apply the forward evolution network $\mathcal{F}_\theta^m$ to perform $K$-step prediction of the system future states, $\hat{\mathbf{u}}(k\Delta) = \mathcal{F}_\theta^{m,(k)}[\mathbf{u}(0)]$, where $k = 1, \cdots, K-1, K, \mathcal{F}_\theta^{m,(k)}$ represents the $k$-fold composition of $\mathcal{F}_\theta^m$:

$$\mathcal{F}_\theta^{m,(k)} = \underbrace{\mathcal{F}_\theta^m \circ \mathcal{F}_\theta^m \circ \cdots \circ \mathcal{F}_\theta^m}_{k-\text{fold}}. \tag{2}$$

After $K$ steps of forward evolution prediction, then, starting with $\hat{\mathbf{u}}(K\Delta)$, we perform $K$ steps of backward evolution prediction using network $\mathcal{G}_\vartheta^m$: $\bar{\mathbf{u}}(k\Delta) = \mathcal{G}_\vartheta^{m,(K-k)}[\hat{\mathbf{u}}(K\Delta)], k = K-1, \cdots, 1, 0$, where

$$\mathcal{G}_\theta^{m,(K-k)} = \underbrace{\mathcal{G}_\theta^m \circ \mathcal{G}_\theta^m \circ \cdots \circ \mathcal{G}_\theta^m}_{(K-k)-\text{fold}} \tag{3}$$

and reach back to time $t = 0$. If the forward and backward evolution networks are both accurate, the forward prediction path and the backward prediction path should match each other. Motivated by this, we define the multi-step reciprocal prediction error for the forward evolution network $\mathcal{F}_\theta^m$ as the deviation between the forward and backward prediction paths:

$$\mathbb{E}[\mathbf{u}(0)] = \sum_{k=0}^{K} \left\| \hat{\mathbf{u}}(k\Delta) - \bar{\mathbf{u}}(k\Delta) \right\|^2. \tag{4}$$

Note that, when computing $\mathbb{E}[\mathbf{u}(0)]$, we only need the current system state $\mathbf{u}(0)$, the forward and backward evolution networks $\mathcal{F}_\theta^m$ and $\mathcal{G}_\vartheta^m$.

**(2) Critical sampling and adaptive evolution operator learning.** In this work, we find that there is a strong correlation between the network modeling error $\mathcal{E}[\mathbf{u}(0)]$ and the multi-step reciprocal prediction error $\mathbb{E}[\mathbf{u}(0)]$. This correlation allows us to predict $\mathcal{E}[\mathbf{u}(0)]$ using $\mathbb{E}[\mathbf{u}(0)]$ which can be computed directly from the current system state without the need to know the ground-truth state. Therefore, once we are able to predict the network modeling error $\mathcal{E}[\mathbf{u}(0)]$ using the multi-step reciprocal prediction error $\mathbb{E}[\mathbf{u}(0)]$, we can develop a critical sampling and adaptive evolution learning algorithm. The central idea is to select samples from locations with large values of error $\mathbb{E}[\mathbf{u}(0)]$ using the following iterative peak finding algorithm. Note that $\mathbf{u}(0) \in \mathbb{R}^n$. Write $\mathbf{u}(0) = [u_1, u_2, \cdots, u_n]$. Let $\mathcal{S}_F^m = \{[\mathbf{u}^j(0) \to \mathbf{u}^j(\Delta) : 1 \leq j \leq J_m\}$ be the current sample set. To determine the locations of new samples, $\{\mathbf{u}^j(0) | J_m + 1 \leq j \leq J_{m+1}\}$, we find the peak value of multi-step reciprocal prediction error $\mathbb{E}[\mathbf{u}(0)]$ at every sampling point $\mathbf{u}(0)$ in the solution space $D$. The corresponding peak location is chosen to be $\mathbf{u}^{J_m+1}(0)$ and the corresponding sample $[\mathbf{u}^{J_m+1}(0) \to \mathbf{u}^{J_m+1}(\Delta)]$ is collected. This process is repeated for $J_{m+1} - J_m$ times to collect $J_{m+1} - J_m$ samples in $\mathbf{\Omega}_m$, which is added to the current sample set:

$$\mathcal{S}_F^{m+1} = \mathcal{S}_F^m \bigcup \mathbf{\Omega}^m = \{[\mathbf{u}^j(0) \to \mathbf{u}^j(\Delta)] : 1 \leq j \leq J_{m+1}\}. \tag{5}$$

Table 1: Samples for learning the system evolution using the baseline method and our method.

| Dynamical System | Baseline | | Our Work | | Ratio |
|---|---|---|---|---|---|
| | Samples | Prediction Error | Samples | Prediction Error | |
| Damped Pendulum | 14400 | $0.02630 \pm 0.01200$ | **417** | **0.02411** $\pm 0.00991$ | 34.53 |
| 2D Nonlinear | 14400 | $0.00037 \pm 0.00021$ | **925** | **0.00035** $\pm 0.00015$ | 15.57 |
| Lorenz System | 1000000 | $0.19685 \pm 0.07768$ | **1765** | **0.19357** $\pm 0.05695$ | 566.57 |
| Viscous Burgers' Eq. | 500000 | $0.01679 \pm 0.00878$ | **19683** | **0.01652** $\pm 0.00818$ | 25.40 |

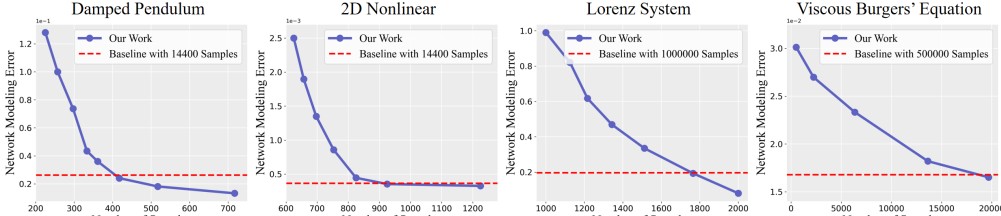

Figure 2: The critical sampling and adaptive learning results on four dynamical systems.

## 3 Experimental Results

### 3.1 Experimental Settings

We consider four representative systems with ODEs and PDEs as their governing equations, as summarized in Table 2 in Appendix. They include (1) the Damped Pendulum ODE equation, (2) a nonlinear ODE equation, (3) the Lorenz system, and (4) the Viscous Burgers' equation (PDE). In Appendix, we provide detailed experimental settings and implementation details.

### 3.2 Performance Results

We choose the evolution learning method developed in [8, 17] as our baseline. On top of this method, we implement our proposed method of critical sampling and adaptive evolution learning. We demonstrate that, to achieve the same modeling error, our method needs much fewer samples.

Table 1 compares the numbers of samples needed for learning the system evolution by the baseline method and our critical sampling and adaptive learning method. The prediction errors are evaluated on 50 arbitrarily chosen solution trajectories in the computational domain. Average errors and standard deviations are reported for each dynamical system. For example, for the Lorenz system, it needs 1,000,000 samples to achieve the modeling error of 0.197. Using our proposed critical sampling method, the number of samples can be reduced to 1,765, while achieving an even smaller modeling error 0.194. The number of samples has been reduced by 567 times. For the Viscous Burgers' PDE system, the number of samples is also reduced by 25 times.

Figure 2 shows the performance comparison results for the four dynamical systems listed in Table 2. In each sub-figure, the horizontal dashed line shows the average network modeling error achieved by the baseline method for the number of samples shown in the legend. This number is empirically chosen since it is needed for the network to achieve a reasonably accurate and robust learning performance. We can see that as more and more samples are selected by our critical sampling method, the network modeling error quickly drops below the average modeling error of the baseline method.

## 4 Conclusion

In this work, we have studied the critical sampling for the adaptive evolution operator learning problem. We have made an interesting finding that the network modeling error is correlated with the multi-step reciprocal prediction error. With this, we are able to perform a dynamic selection of critical samples from regions with high network modeling errors and develop an adaptive sampling-learning method for dynamical systems. Extensive experimental results demonstrate that our method is able to dramatically reduce the number of samples needed for effective learning and accurate prediction of the evolution behaviors.

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

## A  Experimental Settings and Implementation Details

In this section, we provide more details on system configurations and algorithm implementation.

### A.1  Dynamical Systems Studied in this Work

In this paper, we consider four representative systems with ODEs and PDEs as their governing equations. They include (1) the Damped Pendulum ODE in $\mathbb{R}^2$, (2) a nonlinear ODE system in $\mathbb{R}^2$, (3) the Lorenz system (ODE) in $\mathbb{R}^3$, and (4) the Viscous Burgers' equation (PDE). Their governing equations are shown in Table 2.

Table 2: Overview of the 4 governing equation systems we demonstrate in this work.

| SYSTEM | GOVERNING EQUATIONS |
|---|---|
| DAMPED PENDULUM EQUATION | $\begin{cases} \frac{d}{dt}u_1 = u_2, \\ \frac{d}{dt}u_2 = -0.2u_2 - 8.91\sin u_1. \end{cases}$ |
| A 2D NONLINEAR EQUATION | $\begin{cases} \frac{d}{dt}u_1 = u_2 - u_1\left(u_1{}^2 + u_2{}^2 - 1\right), \\ \frac{d}{dt}u_2 = -u_1 - u_2\left(u_1{}^2 + u_2{}^2 - 1\right). \end{cases}$ |
| LORENZ SYSTEM | $\begin{cases} \frac{d}{dt}u_1 = 10\left(u_2 - u_1\right), \\ \frac{d}{dt}u_2 = u_1\left(28 - u_3\right) - u_2, \\ \frac{d}{dt}u_3 = u_1 u_2 - (8/3)u_3. \end{cases}$ |
| VISCOUS BURGERS' EQUATION | $\begin{cases} u_t + \left(\frac{u^2}{2}\right)_x = 0.1u_{xx}, & (x,t) \in (-\pi, \pi) \times \mathbb{R}^+, \\ u(-\pi, t) = u(\pi, t) = 0, & t \in \mathbb{R}^+. \end{cases}$ |

### A.2  System Configurations

For the ODE examples, we follow the procedure in Qin et al. [8] to generate the training data pairs $\{[\mathbf{u}^j(0), \mathbf{u}^j(\Delta)]\}$ as follows. First, we generate $J$ system state vectors $\{\mathbf{u}^j(0)\}_{j=1}^J$ at time 0 based on a uniform distribution over a computational domain $D$. Here, $D$ is the region where we are interested in the solution space. It is typically chosen to be a hypercube prior to the computation, which will be explained in the following. Then, for each $j$, starting from $\mathbf{u}^j(0)$, we solve the true ODEs for a time lag of $\Delta$ using a highly accurate ODE solver to generate $\mathbf{u}^j(\Delta)$. *Notice that, once the data is generated, we assume that the true equations are unknown, and the sampled data pairs are the only known information during the learning process.*

For the first example dynamical system listed in Table 2, its computational domain is $D = [-\pi, \pi] \times [-2\pi, 2\pi]$. We choose $\Delta = 0.1$. For the second system, the computational domain is $D = [-2, 2]^2$. The time lag $\Delta$ is set as 0.1. For the third system, the computational domain is $D = [-25, 25]^2 \times [0, 50]$. The time lag $\Delta$ is set as 0.01.

For the Viscous Burgers' PDE system, because the evolution operator is defined between infinite dimensional spaces, and we approximate it in a modal space, namely, a generalized Fourier space,

in order to reduce the problem to finite dimensions as in Wu and Xiu [17]. We follow the same procedure specified in Wu and Xiu [17] to choose a basis of the finite dimensional space $\mathbb{V}_n$ to represent the solutions, then apply the projection operator to project the snapshot data to $\mathbb{V}_n$ to obtain the training data in the generalized Fourier space. The choice of basis functions is fairly flexible, any basis suitable for spatial approximation of the solution data can be used. Once the basis functions are selected, a projection operator $\mathcal{P}_n : \mathbb{V} \to \mathbb{V}_n$ is applied to obtain the solution in the finite dimensional form.

The approximation space is chosen to be relatively larger as $\mathbb{V}_n = \text{span}\{\sin(jx) : 1 \leqslant j \leqslant n\}$ with $n = 9$. The time lag $\Delta$ is taken as 0.05. The domain $D$ in the modal space is set as $[-1.5, 1.5] \times [-0.5, 0.5] \times [-0.2, 0.2]^2 \times [-0.1, 0.1]^2 \times [-0.05, 0.05]^2 \times [-0.02, 0.02]$, from which we sample the training samples.

Our task is to demonstrate how our proposed method is able to significantly reduce the number of samples needed for evolution learning. Specifically, for the baseline method, we use random sampling, randomly selecting locations in the solution space to collect samples for evolution learning. For example, for the first dynamical system, Damped Pendulum system (ODE) in a 2-D space, the baseline method use 14400 samples to achieve an average network modeling error of 0.026. We then use our method to adaptively discover critical samples and refine the evolution network to reach the same or even smaller network modeling error. We demonstrate that, to achieve the same modeling error, our proposed method needs much fewer samples.

## A.3  Implementation Details

In all examples, we use the recursive ResNet (RS-ResNet) architecture in He et al. [21], Qin et al. [8], which is a block variant of the ResNet and has been proven in Qin et al. [8], Wu and Xiu [17] to be highly suitable for learning flow maps and evolution operators.

For all the 4 systems, the batch size is set as 10. In the two 2-dimensional ODE systems, we use the one-block ResNet method with each block containing 3 hidden layers of equal width of 20 neurons, while in the 3-dimensional ODE system, we use the one-block ResNet method with each block containing 3 hidden layers of equal width of 30 neurons. For the final PDE system, we use the four-block ResNet method with each block containing 3 hidden layers of equal width of 20 neurons. Adam optimizer with betas equal $(0.9, 0.99)$ is used for training. In the two 2-dimensional ODE systems, all the networks are trained with 150 epochs. In the Lorenz system and Viscous Burgers' equation, all the networks are trained with 60 epochs. The initial learning rate is set as $10^{-3}$ , and will decay gradually to $10^{-6}$ during the training process. All networks are trained using PyTorch on one RTX 3060 GPU.

In the four example systems, we evaluate the performance of our models on time duration $t \in [0, 20]$, $t \in [0, 10]$, $t \in [0, 5]$, $t \in [0, 2]$, respectively. For the first two ODE systems, the network modeling error is evaluated by average MSE error at each time step on **50** different arbitrarily chosen solution trajectories. For the Lorenz system, we evaluate the network by average MSE error at each time step under **50** different initial conditions. For the final PDE system, the network modeling error is evaluated by the average $L_2$ norm error on 100 points at time $t = 2$ under **50** different initial conditions.

