# OpenReview forum: "Critical Sampling for Data-Driven Modeling of Unknown Dynamical Systems"
_robot-learning.org/CoRL/2023/Workshop/OOD — OOD Workshop @ CoRL 2023_

### Official Review · Reviewer_xYBo · 2023-10-16
**Weak accept because only tangentially related to workshop topic**

**Rating:** 6
**Confidence:** 4

**Review:**

This paper presents an adaptive sampling-based learning method for dynamical systems, e.g., those represented by an ODE or PDE. Their method aims to learn a model of the evolution operator, i.e., how states transition, that can accurately predict future evolution while using the minimum number of observational samples. Samples can be judiciously chosen by using iterative peak finding to identify points with the highest network modeling error. While this error cannot be directly estimated, the key insight in this paper is that the model error is closely correlated with the "multi-step reciprocal prediction error", i.e., the error between a state at time t' generated by forward propagation and a state at the same time t' generated by backward propagation, which encodes a notion of "goodness" of the dynamics model. They show that their method is able to identify these critical samples online in a streaming fashion and is able to learn a good evolution operator in a fraction of the number of samples required by the benchmarks.

This paper is well-written and the authors well motivate their choice of error estimation. The approach is highly effective for the chosen problem domain However, its relevance to the OOD workshop is debatable. On the one hand, identifying critical sample points, where model error is highest online during deployment is a key component of adapting the model through discovery, potentially in OOD scenarios. On the other hand, their method relies on accurate forward and backward dynamics models, and is tested on systems that are modeled with ODEs or PDEs where the input distribution remains the same.

While the paper does not directly address the OOD problem, I believe some of these ideas can be extended to sampling for dynamics learning in the presence of OOD. I recommend a week accept.

---

### Official Review · Reviewer_duqT · 2023-10-16
**Review of OOD Workshop Submission 11**

**Rating:** 7
**Confidence:** 5

**Review:**

Thanks for submitting to this workshop!  This paper presents an idea about how to efficiently sample data points for accurately learning an unknown dynamical system.  To take more samples from areas in the state space where the dynamics are less well known, one first needs a way to assess where the dynamics are less well known.  The authors propose to learn two neural networks from sampled data:  one which predicts forwards and one backwards.  Then, if you run the system forward a few steps, then backwards the same number of steps, you should end up where you started.  If you don't, then you don't know the dynamics well at this state and need to sample more and relearn.  The authors show it works on some classical 2 and 3 D nonlinear systems, including a PDE.  I think this idea is interesting and would like to see it discussed more.  Some questions I have are:
 -- Works for time-invariant dynamics, not time varying?
 -- Works for systems where you can run it forwards and backwards -- that is, if it is a control system you need to be able to run the control backwards?

---

### Decision · Program_Chairs · 2023-10-17

**Decision:**

Accept

**Comment:**

We agree with the reviewers’ assessment that this work is technically sound and will contribute to productive, topical discussions at the 2023 Workshop on OOD Generalization in Robotics. In particular, we appreciate that the method applies to improving a model when it experiences novel inputs on which the current performance is inaccurate. We agree with the reviewers’ comments that the impact of your work (in the context of this workshop) would be improved by highlighting specifically the performance of the method when new (OOD) regions of the state space are explored e.g., due to varying input distributions. We recommend the authors incorporate the reviewers’ feedback into their camera-ready submission to further improve their manuscript.